# Proteome remodeling in the zoospore-to-vegetative cell transition of the stramenopile *Aurantiochytrium limacinum* reveals candidate ectoplasmic network proteins

Alejandro Gil-Gomez[1], Ben Leyland[2], Anbarasu Karthikaichamy[2¤], Rebecca C. Adikes[3], David Q. Matus[4], Joshua S. Rest[1], Jackie L. Collier[2]*

**1** Department of Ecology and Evolution, Stony Brook University, Stony Brook, New York, United States of America, **2** School of Marine and Atmospheric Sciences, Stony Brook University, Stony Brook, New York, United States of America, **3** Department of Biology, Siena College, Loudonville, New York, United States of America, **4** Department of Molecular and Cell Biology, University of California at Berkeley, Berkeley, California, United States of America

¤ Current address: Phycoworks, 58 Wood Lane, London, United Kingdom W12 7RZ
* jackie.collier@stonybrook.edu

## Abstract

Thraustochytrids are marine protists of ecological and biotechnological importance. Like many other eukaryotes, their life cycle includes a critical transition from a flagellated, swimming zoospore dispersal stage to a settled, surface-attached, growing vegetative cell. Unlike other eukaryotes, the settling vegetative cells of thraustochytrids (and their labyrinthulomycete relatives) attach to surfaces by producing a unique structure known as the ectoplasmic network, and its associated connection to the cytoplasm, the bothrosome. We conducted time-course proteomics and microscopy to study this transition in the model thraustochytrid *Aurantiochytrium limacinum* ATCC MYA-1381. We identified 623 proteins significantly differentially expressed between zoospores and samples collected 2, 4, 6, and 8 hours after settlement. Analysis of the differentially expressed proteins revealed broad cellular changes during the transition from zoospore to vegetative cell, including shifts in motility, signaling, and metabolism. A relative enrichment of proteasomal and ribosomal components in the zoospores suggests these proteins are stockpiled, priming the zoospore for rapid protein turnover upon settlement. Flagellar proteins were strongly downregulated upon settlement, coinciding with loss of motility. Environmental sensing systems, such as channelrhodopsins, declined post-settlement. The proteomic changes also suggest that zoospores rely on catabolism of stored lipids by beta-oxidation, whereas settled vegetative cells shift towards anabolic metabolism, including gluconeogenesis (growth media contained glycerol), and the biosynthesis of membrane lipids, amino acids, and nucleic acids. A search for proteins which were upregulated during vegetative cell settlement, and which were phylogenetically divergent in thraustochytrids,

**Data availability statement:** Most relevant data are within the manuscript and its Supporting Information files. The proteomics files were too large, so those data will be available at dryad https://doi.org/10.5061/dryad.2z34tmpxj.

**Funding:** This work was supported by Gordon and Betty Moore Foundation (https://www.moore.org/) Grant GBMF4982 to J.L.C. and J.S.R. by a Seed Grant 2020F 246520 from the Stony Brook University School of Marine and Atmospheric Sciences (https://www.stony-brook.edu/somas/) to J.L.C., J.S.R. and D.Q.M. The funders had no role in study design, data collection and analysis, decision to publish, or preparation of the manuscript.

**Competing interests:** The authors have declared that no competing interests exist.

yielded a list of potential ectoplasmic network or bothrosome candidates, including potential homologs of micronemal adhesins and membrane-trafficking proteins. Our findings illuminate a critical life-history transition in *A. limacinum*, and identify targets for understanding the evolutionary origins and functions of unique labyrinthulomycete structures.

## Introduction

The ecology and life history of many organisms converge at a pivotal point: the transition from dispersing and seeking out new habitats to settling down and growing. In many protists, the dispersal stage is a flagellated zoospore specialized to locate appropriate habitat, and equipped with endogenous reserves to provide both energy for motility and building blocks for rapid differentiation into the growth stage [1]. The transition from zoospore to growing (vegetative) cell has been investigated in detail in a few lineages, seeking insights to targets for pest management or population management for exploited species; examples include zoosporic fungi such as chytrids and blastocladids [2–5], and some stramenopiles, including kelps [6] and oomycetes [7–9]. Zoospore proteomes are generally relatively enriched in proteins involved in energy generation, motility, and signaling, while vegetative cells are generally relatively enriched in proteins involved in nutrient acquisition, metabolism, and cell proliferation [10,11]. Because of their unique biology and genetic tractability [12], thraustochytrids represent a compelling additional model for studying the metabolic and morphological changes underlying habitat colonization by protists.

Thraustochytrids (members of the stramenopile phylum Labyrinthulomycota; [13]) are fungus-like osmoheterotrophic marine protists that are of both ecological and economic importance [14]. Thraustochytrids have biflagellate swimming zoospores typical of stramenopiles, with a tinsel-type flagella oriented anteriorly and a whiplash-type flagella oriented posteriorly [15]. Thraustochytrid zoospores can swim actively for hours, probably using lipid reserves, at speeds ~100 micrometers per second, exhibiting chemotactic and phototactic behaviors before settling on a suitable substrate [16–19]. The settled zoospore undergoes a major change in morphology and physiology as it transforms to a vegetative cell. The vegetative cell will subsequently grow and develop into a zoosporangium, which can produce the next generation of zoospores. Thraustochytrid vegetative growth includes production and accumulation of substantial lipid reserves rich in essential long-chain polyunsaturated fatty acids (LC-PUFA), which may be utilized in the zoospore stage or during the early transition to vegetative growth [20–23].

One feature unique to the zoospore-to-vegetative cell transition in thraustochytrids (and other labyrinthulomycetes) is the production of the ectoplasmic network (EN): branched extensions of the plasma membrane which have an actin cytoskeleton and are involved in the search for and attachment to food sources by vegetative cells [24,25]. In the thraustochytrid *Schizochytrium aggregatum*, the EN arises from the basal side of the settled zoospore cell body, connected by a single bothrosome

formed *de novo* near the site of the flagellar microtubular roots [26]. The EN seems functionally analogous to the rhizoids of chytrid fungi [27] or to animal filopodia; however, the EN may represent an independent instance of the evolution of surface attachment and feeding. A first step towards elucidating its evolutionary origin, and relationships to other systems is identifying the proteins in the EN and bothrosome. Understanding their molecular composition will also provide insights into their function in substrate attachment and nutrient acquisition.

Here, we combine video microscopy and time course proteomics to characterize protein expression dynamics during the transition from zoospore to settled vegetative cells with an EN. We studied this in *Aurantiochytrium limacinum* ATCC MYA-1381 [28], a model strain for thraustochytrids with expanding genomic and functional genomic tools [12,29,30]. In doing so, we focus on two goals (1) characterizing the broad physiological changes in the transition and (2) identifying putative protein components of the bothrosome and EN.

## Results and discussion

We characterized *A. limacinum* zoospore settlement and EN production using high-definition time-lapse microscopy (Fig 1, S1 File). Zoospores remained motile for less than an hour, and flagella remained visibly attached to settled cells for ~ 90 min, after which settled cells were firmly attached to the substrate. EN development was visible after 4–6 hours, with well-developed EN networks by 8 hours. Cell bodies were clearly growing larger by 2.5–3 hours, and cell division

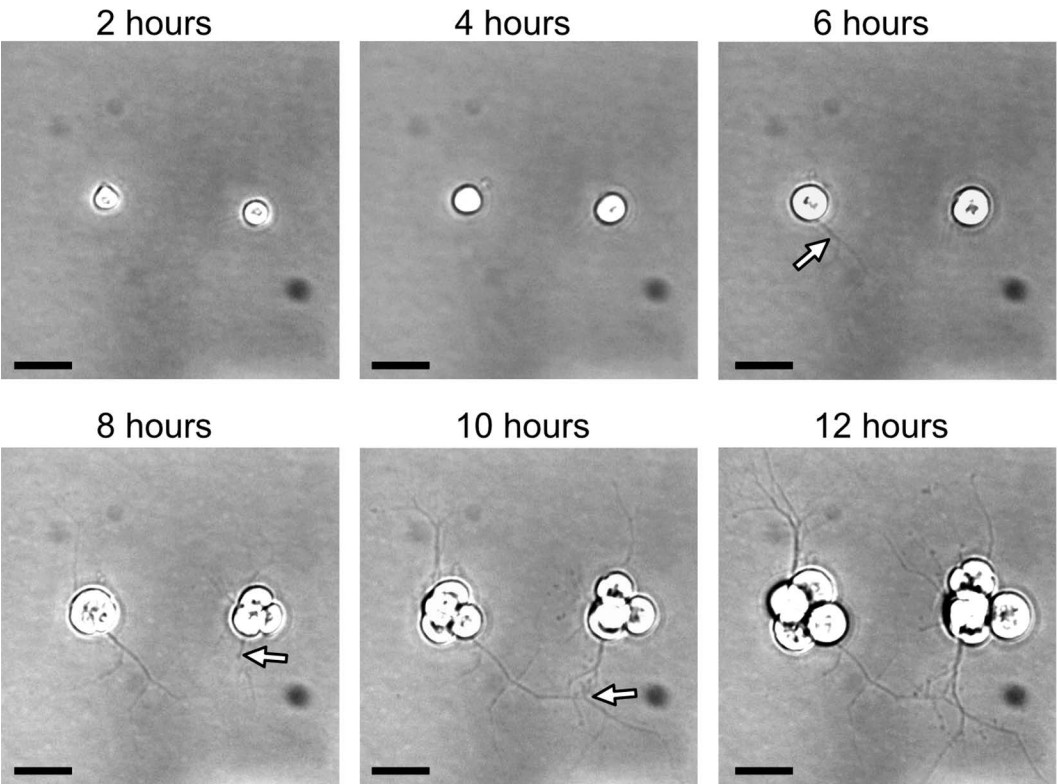

**Fig 1. Time-resolved high-definition imaging of *A. limacinum* reveals coordinated ectoplasmic network (EN) development and cell division following zoospore settlement.** Key events include: initial zoospore motility, flagellar attachment (~90 min post-settlement), visible EN development (4-6 hours), and robust EN network formation by 8 hours. Arrows point out the emergence, extension, and connection of some strands of the EN. Growth in cell body size is noticeable at 2.5-3 hours, leading to cell division between 6-8 hours. Black scale bar at lower left indicates 10 μm. The contrast and brightness of these images was adjusted to increase the visibility of the EN; original un-adjusted images are available in S1 File.

was detected after 6–8 hours (S1A Table). This timing is generally consistent with the report of Morita et al. [20], who documented synchronous growth for up to 8 hours after zoospore settlement, with nuclear divisions beginning toward the end of that time, and cell divisions beginning soon after.

Across five time points (zoospores at T0 and settled cells after 2, 4, 6, or 8 hours), the 3783 predicted proteins detected in our proteomic analysis ranged 4302-fold in abundance, between 5299 and 22799791 intensity units (log$_2$ intensities 11.8 to 24.4) (S2 Table). After variance-stabilizing normalization, the distribution of protein abundance was similar across all samples (S1A Fig). The 20 most abundant proteins detected across all samples (S1B Table) predominantly play fundamental roles in core cellular processes, including components of the actin and tubulin cytoskeleton, translation elongation factors, components of the F0F1-type ATP synthase, and enzymes in central metabolic processes, including purine and pyrimidine biosynthesis, amino acid metabolism, glycolysis/gluconeogenesis, and the TCA cycle. Notably, the 9th most abundant protein (A41691, KOG1429; identifiers such as A### refer to proteins in the JGI Aurli1 genome annotation; see Methods) stood out due to its putative dual function as a dTDP-glucose 4,6-dehydratase and UDP-glucuronic acid decarboxylase. These enzymatic activities are involved in the biosynthesis of precursors (dTDP-rhamnose and UDP-xylose) used in polysaccharide synthesis. Given that labyrinthulomycete vegetative cells have cell coverings comprising Golgi-derived polysaccharide scales, and extracellular polysaccharide matrices composed of various sugars (e.g., glucose, galactose, xylose, fucose, mannose, rhamnose) [31,32], the high abundance of A41691 underscores its potential importance for cell wall biogenesis in *A. limacinum*.

The most abundant proteins detected reflect the metabolism of *A. limacinum* vegetative cells, which are specialized for growth and biosynthesis. These proteins are primarily involved in protein synthesis, energy production, and core metabolic pathways. This pattern is also consistent with previous studies, which found that translation factors, glycolytic enzymes, and ATP synthase are among the most abundant cellular proteins [33].

While the expression of most proteins remained relatively stable (mode and mean log fold-change very close to 0, S1B Fig), some underwent consistent and marked expression changes. The greatest decline was −1.75 log2-fold-change for Hypothetical Protein FCC1311_091412 (A32851) at T4, and the greatest increase was 1.81 log2-fold-change for Mitochondrial F1F0-ATP synthase subunit delta/ATP16 (A68446) also at T4. Principal Component Analysis (PCA) of all detected proteins highlighted two main axes of variation that correlated with early settlement and subsequent vegetative development, with PC1 capturing 62.2% and PC2 capturing 16.6% of the total variance (Fig 2). From time 0 (zoospores) to 4 hours post-settlement, the samples moved from left to right along PC1, suggesting that this axis is associated with changes over the first 4 hours of development from zoospore to vegetative cell. From time 0–2 hours post-settlement, the samples moved from top to bottom of PC2, then from 2 to 8 hours post-settlement they moved back up PC2, suggesting that this axis may capture transient changes associated with settlement and early development of vegetative cells, which may include loss of proteins associated with flagellar motility as well as with EN and bothrosome development.

623 of the 3783 detected proteins were significantly differentially expressed at least at one time point in comparison to zoospores (time 0) (S2 Table, S2 and S3 Figs; see Methods). A total of 356 proteins showed a greater relative abundance compared to zoospores at one or more time points, which we will refer to as 'upregulated'. Conversely, 267 proteins were found to be at a lesser relative abundance than in zoospores at one or more time points, and these will be referred to as 'downregulated'. Consistent with the PCA, the largest single group of upregulated proteins, 141 (40%), were exclusively upregulated at time point 2, whereas the largest single group of downregulated proteins, 87 (33%), were downregulated across all time points (S3 Fig). Among the coordinated changes expected in the transition from zoospore to vegetative cell would be loss of flagellar proteins, which is illustrated by the significant reduction in two alpha-tubulin proteins (A0A6S-8DQS1 and A146288, both cluster C1; clusters described below) and the putative mastigoneme protein A117061 (also cluster C1), across multiple time points (Fig 3).

We identified four distinct patterns of protein expression over the settlement time course by grouping the 623 significant proteins into four clusters (C1 to C4) using k-means and Pearson distance, and visualized them in a heatmap of log2

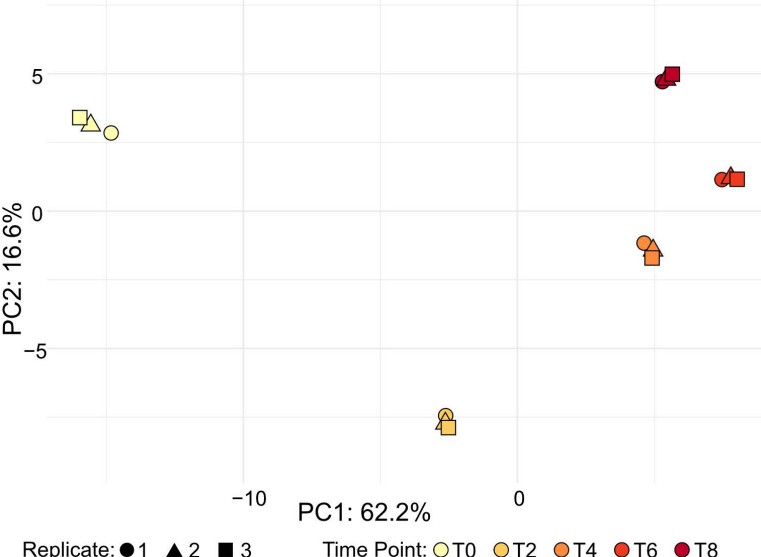

**Fig 2. Principal Component Analysis reveals major axes of protein expression variation during settlement and early vegetative growth.** PC1 (x-axis, 62.2% of variance) represents developmental changes from zoospores to vegetative cells over the initial 4 hours. PC2 (y-axis, 16.6% of variance) reflects transient changes associated with settlement and early vegetative cell development, particularly from 0 to 2 hours post-settlement, including loss of flagellar proteins and EN/bothrosome development. Data points are color-coded by time point, highlighting consistency across replicates.

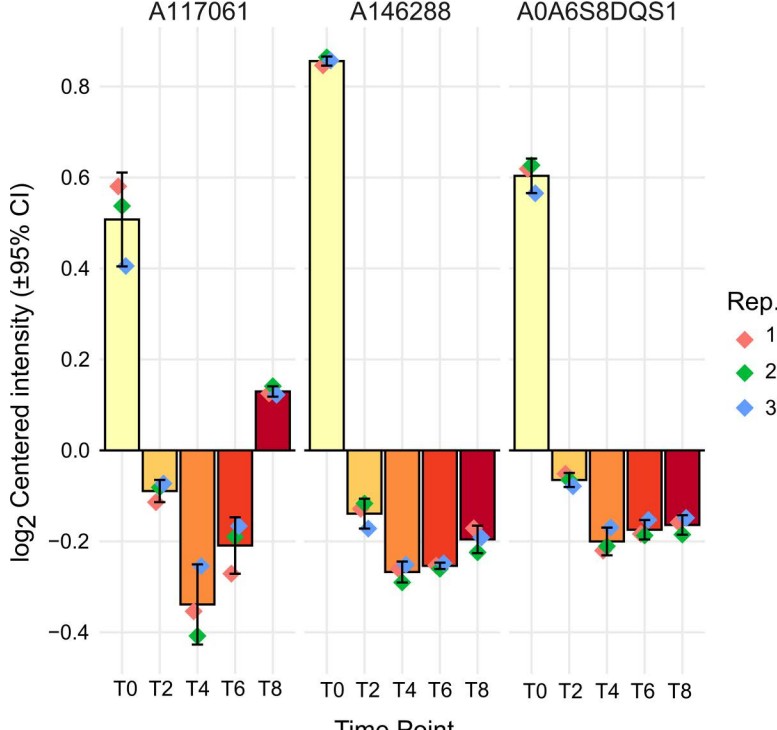

**Fig 3. Temporal expression patterns of some predicted flagellar proteins during the transition from free-swimming zoospore to non-motile vegetative cell.** The figure displays the log-centered intensity of two predicted alpha-tubulin proteins (A0A6S8DQS1 and A146288) and the putative mastigoneme protein A117061 across five time points: T0, T2, T4, T6, and T8. 'Rep.' indicates independent biological replicates. The reduction in expression levels over time suggests a decline in structures associated with motility, consistent with the observed loss of flagella and related proteins in the transition.

centered intensity values (Fig 4). Consistent with the PCA, replicates of the same time points clustered together, T6 and T8 clustered together, and these clustered with T4. Clusters C1 and C2 contained proteins that were in relatively high abundance at time point T0 (in zoospores) compared to later time points. For example, cluster C1 includes the alpha-tubulin and mastigoneme proteins shown in Fig 3. About 90% of the proteins from cluster C1 (82/91 proteins) and 51% from cluster C2 (91/177) were significantly downregulated at T2, and the majority (198, 74%) of proteins from clusters C1 (64/91) and C2 (134/177) were significantly downregulated at T8 (Fig 4, S5 Fig). Cluster C4 contained proteins that were in relatively low abundance in zoospores, and nearly all (191/195) significantly increased in abundance during early settlement (T2). Cluster C3 contained proteins that were in relatively low abundance in zoospores and nearly all (155/160 proteins) were significantly upregulated at time points T6 and/or T8. Expression data for all proteins can be accessed in S2 Table.

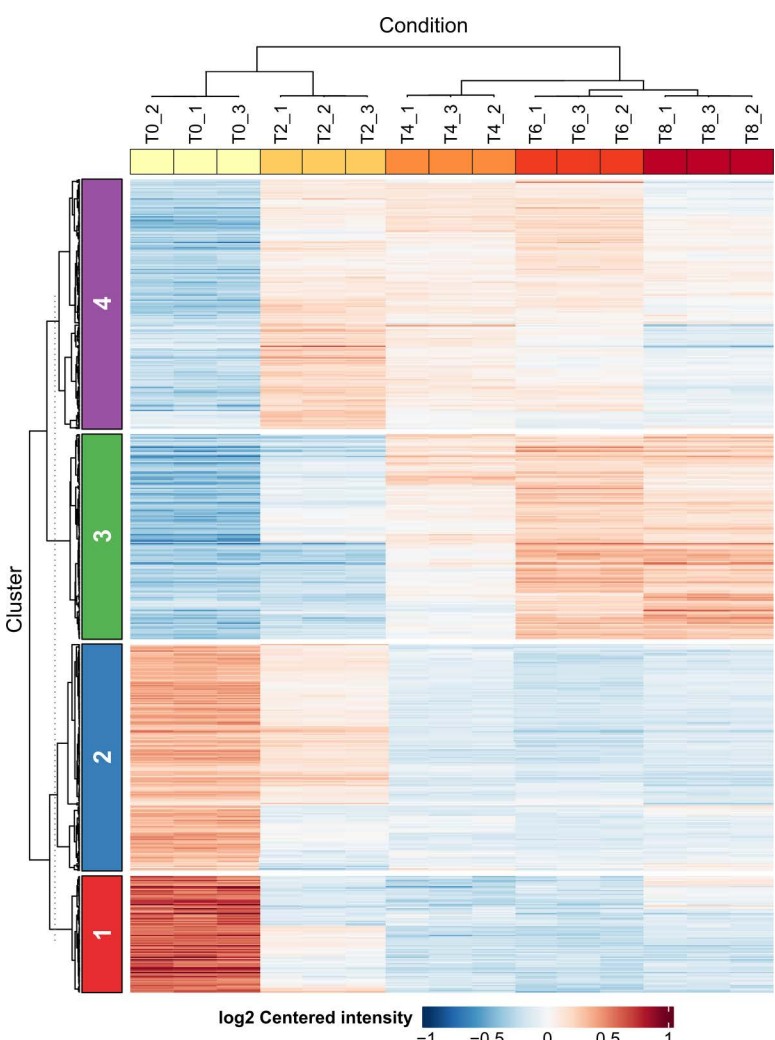

**Fig 4. Heatmap visualization of four temporal expression clusters of 623 significant proteins revealing dynamic changes across the settlement time course.** Proteins were clustered using k-means and Pearson distance across time points T0, T2, T4, T6, T8 and replicates _1, _2, and _3. Colors represent log2-centered intensities (i.e., average log2 fold-change scaled protein-wise). Gene clusters C1-C4 are indicated at left.

## Functional remodeling in the zoospore to vegetative cell transition

To explore the changes in cellular structure and function as vegetative cells developed from zoospores, we tested which KOG classes (KOG--) or KEGG BRITE (ko---) groups were over- or underrepresented among the significantly differentially expressed proteins. S6 Fig and S1D Table display the full set of significantly enriched KOG/ko groups, including enrichment tests for each individual expression cluster (C1-C4). A curated subset of enrichment tests and KOG/ko groups of interest is shown in Fig 5. Gene Ontology (GO) enrichment was also performed for the cluster-based analysis (S1D Table).

## Transition away from motility and environmental sensing

One of the most obvious morphological changes during the transition from free-swimming zoospore to non-motile vegetative cell is the loss of structures associated with flagellar motility. Consistent with the changes observed for alpha-tubulin (Fig 4), we observed a notable reduction in proteins linked with motility. Specifically, cilia and flagella-related proteins (such as ko03037 in Fig 5) and cytoskeleton proteins (such as ko04812 in Fig 5) were downregulated during the transition from zoospores to vegetative cells, likely associated with the loss of flagella during this phase.

KOG 'signal transduction mechanisms', which includes protein kinases (ko01001) and other pathways, was significantly overrepresented in early-downregulated cluster C1 and the combined set of downregulated proteins (C1+C2) (Fig 5, S6 Fig). This is consistent with a decline in the environmental sensing systems needed by zoospores but not by vegetative cells. For example, the 'RubyACR' anion-conducting channelrhodopsin A35957 (called AlACR1 in [34]) was significantly downregulated in all vegetative cell time points compared to zoospores. Other channelrhodopsins, A7690 and A143491 (AlACR2 and AlACR3 in [34], respectively) were also downregulated, though not significantly (S2 Table). Notably,

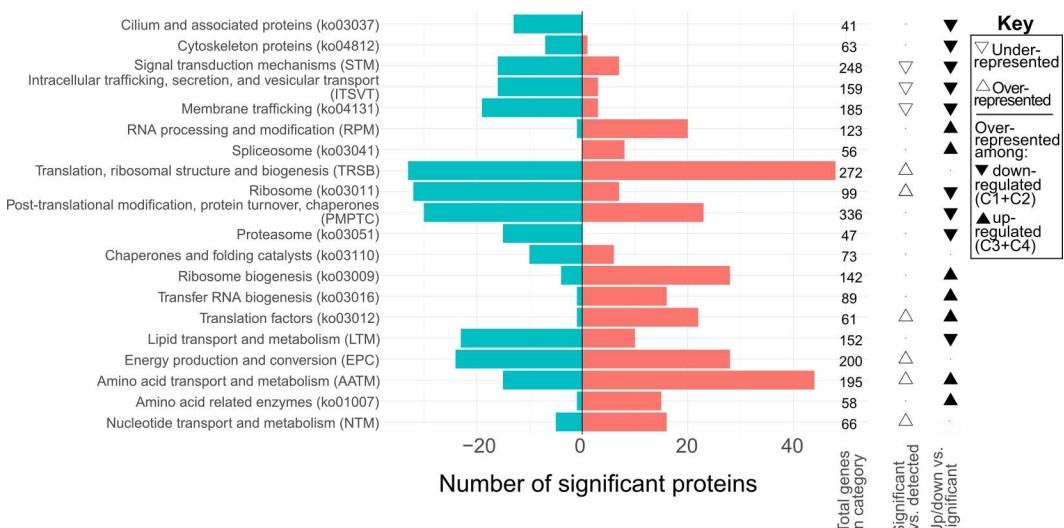

**Fig 5. Functional classification of differentially expressed proteins during the transition from zoospores to vegetative cells in *A. limacinum*.** Bars show the total number of significantly upregulated (red, right-facing) and downregulated (blue, left-facing) proteins (at any time point) relative to zoospores (time 0) for each selected KOG class or ko group. Unfilled triangles in the first column at right indicate functional groups significantly overrepresented (upward triangle) or underrepresented (downward triangle) among all differentially expressed proteins (Fisher's Exact Test with 5% false discovery rate). Filled triangles in the last column indicate functional groups overrepresented among upregulated (C3 and C4, upward triangle) or downregulated (C1 and C2, downward triangle) genes in vegetative cells in comparison to zoospores. The comparisons shown here represent a curated subset of the full enrichment analysis shown in S6 Fig.

however, the signal transduction category was underrepresented among all differentially expressed proteins overall (Fig 5), suggesting that most signal transduction pathways remain relatively stable during the transition.

Several categories related to intracellular trafficking were also overrepresented among downregulated genes in the transition from zoospore to growing vegetative cell (KOG class 'intracellular trafficking, secretion, and vesicular transport'; membrane trafficking ko04131; vesicle-mediated transport GO:0016192; intracellular protein protein transport GO:0006886; Fig 5 and S1D Table). Declines in specific proteins like the coatomer COPI subunits (A71350 and A142887 in cluster C1, A68400 and A139751 in cluster C2) are consistent with an enrichment of zoospores in vesicle-related activities involved in their initial attachment to surfaces [26]. These KOG/ko categories were also underrepresented among all differentially expressed proteins, again suggesting overall stability during this transition (Fig 5).

### Remodeling of gene expression machinery

Changes in proteins associated with gene expression machinery suggest the zoospore could be poised for rapid proteome remodeling, both in the production of new proteins (relatively high ribosome content), and degradation of existing proteins (relatively high proteasome content). While we did not detect significant changes in ko groups, KOG classes, or GO terms associated with transcription, KOG class 'RNA processing and modification' and spliceosome proteins (ko03041) were overrepresented among upregulated proteins (Fig 5), particularly cluster C4 (S6 Fig). Proteins associated with translation, protein maturation, and protein trafficking showed strong responses. Ribosome (ko03011) proteins, along with KOG class 'post-translational modification, protein turnover, chaperones', proteasome (ko03051), and the GO terms ribosome (GO:0003735) and structural constituent of ribosome (GO:00037335), were downregulated in vegetative cells compared to zoospores, particularly in cluster C2 (Fig 5, S1D Table, S8A and S8B Figs). Chaperones and folding catalysts (ko03110) were also overrepresented in cluster C2, and several GO terms associated with protein folding and turnover (such as GO:0005839, proteasome complex, and GO:0006457, protein folding) were enriched among downregulated genes. Conversely, proteins involved in ribosomal biogenesis (ko03009), tRNA biogenesis (ko03016), and translation factors (ko03012), along with GO terms related to translation initiation and elongation, were enriched in genes upregulated in vegetative cells compared to zoospores, particularly in cluster C4 (Fig 5, S1D Table, S7 Fig).

It is known that the very early (first ~30 minutes) steps of zoospore germination in the chytrid *Blastocladiella emersonii* do not require protein or RNA synthesis; chytrid zoospores have stocked mRNA and proteins in the nuclear cap that support a bout of protein synthesis essential for completion of germination [35]. A structure like the nuclear cap has not been reported in labyrinthulomycete zoospores, but our data suggests they may be similarly pre-programmed for early steps of settlement and transformation, as has also been observed in some oomycetes [7]. Comparison with published RNA-seq data from *A. limacinum* zoospores is consistent with protein stockpiling (see Conclusions, below).

### Metabolic reprogramming supports cell growth

Oxidation of endogenous fatty acids is commonly reported to be an important energy source for zoospores [36–38]. Many organisms, including many stramenopiles, store fatty acids in the form of triacylglycerols and sterol esters, sequestered in specialized organelles, called lipid droplets, which can serve as an energy reserve [39–41]. Thraustochytrid zoospores are commonly, though not universally, reported to contain lipid droplets [42]. Morita et al. [20] observed a decline in the number of lipid droplets in the first 4 hours after zoospore settlement and reported that the zoospores had much greater content of sterol esters (as % total fatty acid) than vegetative cells.

Consistent with this shift away from lipid-based energy reserves, proteins involved in lipid transport, storage, and metabolism were significantly downregulated during the transition to vegetative cells. This includes KOG class 'lipid transport and metabolism' (LTM) (Fig 5) and several GO terms associated with fatty acid metabolism (S1D Table). Three proteins associated with the lipid droplet [43] were significantly downregulated (acyl-CoA synthetase, A142380; a Rab1/YPT1 GTPase, A77077; short-chain acyl-CoA dehydrogenase, A115689; all in cluster C2). The downregulation of peroxisomal

beta-oxidation enzymes such as enoyl-CoA hydratase (HSD17B4, A11390, C2) and 3-hydroxyacyl-CoA dehydrogenase (fadN, A41946, C2), along with mitochondrial lipid metabolism proteins such as carnitine/acylcarnitine translocase (CACT, A46635, C2), short-chain 2-methylacyl-CoA dehydrogenase (ACADSB, A116973, C2), trifunctional enzyme subunit alpha (HADHA, A136855, C1), carnitine O-palmitoyltransferase 2 (CPT2, A138285, C2), and acyl-CoA dehydrogenase (ACADM, A142917, C2), suggest a shift away from the breakdown of stored triglycerides and oxidation of long-chain fatty acids in the peroxisome and mitochondria during the transition to a vegetative life-stage ([44]; S8C Fig). The first step of peroxisomal fatty acid beta-oxidation produces $H_2O_2$, and the decline of a peroxisomal catalase (A138555, C1) would be consistent with a decrease in $H_2O_2$ production by that pathway. Also downregulated was the mitochondrial electron-transfer flavoprotein ubiquinone oxidoreductase (ETFQO, A46910, C2), identified as essential for mitochondrial fatty acid beta-oxidation and branched-chain amino acid degradation [45].

At the same time, components of the lipid transport and metabolism class linked to anabolic biosynthesis, such as isoprenoid biosynthesis, were upregulated during the transition. These include the mevalonate isoprenoid biosynthetic pathway (GO:0008299; S8D Fig, S1D Table) and related reactions, including two putative sterol 24-C-methyltransferases (A40066, C4 and A136674, C3; both GO:0003838), and another methyltransferase (A43854, C4) previously reported to be associated with the lipid droplet [43].

Carbohydrate metabolism also underwent substantial reprogramming, shifting from glycolysis and TCA cycle activity in zoospores toward gluconeogenesis in vegetative cells. Mitochondria-associated phosphoenolpyruvate carboxylase (A46884, C2), and GO terms associated with glycolysis (such as GO:0006096, including phosphofructokinase A139427, C1; S1D Table, S8E Fig), were more abundant in zoospores. In contrast, the cytoplasmic first committed step of gluconeogenesis, phosphoenolpyruvate carboxykinase (A84665, C3), along with fructose bisphosphatase (A70359, C4), was more abundant in vegetative cells. Also more abundant in the zoospores were several proteins associated with the TCA cycle, including pyruvate dehydrogenase (GO:0004739, A46525 and A117291 in C2), acetyl-CoA synthase (A83581, C1), citrate synthase (A36111, C2), and 2-oxoglutarate dehydrogenase (A142771, C2). On the other hand, malate dehydrogenase (A79000, C3) was more abundant in vegetative cells (S8F Fig). This pattern may reflect both a developmental transition and adaptation to the growth medium, which contains glycerol rather than glucose, along with the glucogenic amino acid glutamate.

The metabolic reprogramming extended to pathways supporting active cell growth and reproduction, marked by upregulation of amino acid and nucleotide metabolism. This included upregulation of the KOG classes 'amino acid transport and metabolism' and 'nucleotide transport and metabolism', the BRITE group amino acid related enzymes (ko01007), and several related GO terms (S8G and S8H Figs, S1D Table). Several enzymes in one-carbon metabolism were also upregulated (S8I Fig). The upregulation of proliferating cell nuclear antigen (PCNA, A115200, C3) by T2 could indicate an early onset of DNA replication. Yet many basic components of cell structure and function, such as ubiquitin system (ko4121), DNA repair and recombination proteins (ko03400), and KOG class 'nuclear structure', were underrepresented among significant proteins, suggesting that the relative protein investment in these systems is not much affected by the transition from zoospore to vegetative cell.

Finally, changes in energy production systems indicated a shift in the way ATP synthesis and regeneration are balanced during vegetative cell growth. KOG class 'Energy production and conversion' is overrepresented among differentially expressed proteins, with a similar number upregulated and downregulated (Fig 5; S1C Table). Several components of F-type ATPase (GO:0046933) were upregulated, while components of V-type ATPase were downregulated (S8J Fig). Also downregulated in vegetative cells were all three predicted ATP:guanido phosphotransferases (A42973, C2; A45810, C1; A140607, C2; GO:0004054), which are components of the phosphotaurocyamine pathway that may play a role in providing ATP to the flagella [46]. Adenylate kinases may also help replenish ATP [46]. Among eight putative adenylate kinases predicted in the *A. limacinum* genome, seven were detected in the proteome, and only one (A65144, C2) was differentially expressed, being more abundant in the zoospore stage (T0).

Taken together, these results indicate that the transition from zoospore to vegetative cell is characterized by coordinated metabolic reprogramming–from lipid catabolism to anabolic processes that support sustained cell growth.

## Candidate proteins for bothrosome and ectoplasmic network

The bothrosome is a membranous organelle ~200 nm in diameter, located at a cup-like plasma membrane invagination. The basal layer of electron-dense material of the bothrosome is connected on one side with the cytoplasm and endoplasmic reticulum, and on the other with the interior of the EN [47,48]. The EN is a wall-less, membrane-bound compartment that may contain 'net elements' described as vesicles or internal membrane cisternae, but no cytoplasmic organelles like ribosomes or mitochondria [49]. New EN is thin, but thickens and develops more complex internal net elements after contacting a food particle and is actively involved in the hydrolysis of substrates and uptake of nutrients [24,25].

Because the EN and bothrosome are morphologically unique to Labyrinthulomycota, the core genes involved in orchestrating the EN/bothrosome likely include both widely conserved proteins (e.g., actin) and potentially novel factors specific to Labyrinthulomycota. We hypothesized that proteins uniquely conserved in Labyrinthulomycota and upregulated during EN formation are promising candidates for bothrosome and EN functions. Specifically, we looked for candidate proteins which are upregulated during the transition from zoospores to the EN-producing vegetative cells, and which have identifiable homologs in other Labyrinthulomycota, but no high-scoring homologues in other Eukaryota. In total, 18 such genes were identified, listed in Table 1 and S1E Table. None have been experimentally tested, and most lack functional annotations by standard sequence similarity methods. To glean insights into their potential roles, we analyzed these proteins using "deep homology" approaches (Phyre2 and HHPred; see methods; note that candidates do not have recognizable homology outside labyrinthulomycetes using "standard" homology searches such as BLAST). The 18 proteins could be divided into three categories: 1) proteins likely performing general cellular functions (e.g., transcription, translation, metabolism) that are upregulated during vegetative growth, and may be unusually divergent in Labyrinthulomycota, but are less likely to be involved in the EN/bothrosome; 2) proteins with features suggestive of direct EN/bothrosome involvement; and 3) proteins too divergent to assign confident structure or function. In particular, we are interested in candidates that fall into the latter two categories.

**Table 1. Potential EN/bothrosome candidate proteins.**

|  | ID | Possible Structure/Function | Cluster |
|---|---|---|---|
| **Putative EN-Related Candidates** | A0A6S8AHQ2 | Extracellular microneme | 3 |
|  | A142921 | type VI secretion protein? | 3 |
|  | A42866 | TRAPPC11/BAR/Endophilin? | 3 |
|  | A47522 | SXP/RAL-2 family protein? | 3 |
|  | A141237 | Late embryogenesis abundant domain-containing protein/ Apolipoprotein-like? | 3 |
|  | A43122 | ? | 3 |
|  | A442 | ? | 4 |
|  | A46070 | ? | 4 |
| **Candidates with Likely General Cellular Functions** | A140971 | Mitoribosome | 4 |
|  | A0A6S8CF23 | Mitochondrial cytochrome? | 4 |
|  | A0A6S8D2A7 | Monothiol glutaredoxin-S11? | 3 |
|  | A139465 | Polyadenylate-binding protein | 4 |
|  | A46158 | ATP synthase (inhibitor?) | 3 |
|  | A139280 | Translation initiation factor | 4 |
|  | A141478 | RNA helicase | 3 |
|  | A44843 | Thiol-disulfide oxidoreductase | 3 |
|  | A41474 | Ribosomal protein? | 4 |
|  | A44457 | 3-dehydroquinate dehydratase | 3 |

See S1E Table for more details. Annotations with a '?' indicate uncertainty of prediction (<90% confidence).

Three candidate proteins (A0A6S8AHQ2, A142921, and A42866) have bioinformatic predictions suggestive of direct involvement in EN formation or function. A0A6S8AHQ2 exhibits some similarity to micronemal proteins from parasitic protists. Phyre2 suggested A0A6S8AHQ2 matches micronemal protein 4 from *Toxoplasma gondii* (94.5% confidence) and SML-2 from *Sarcocystis muris* (93.8% confidence) (Table 1 and S1E Table). HHPred similarly identified micronemal protein 4 as its highest-confidence match (probability 95.5%). Microneme proteins are adhesins, which function in host cell recognition, attachment, and penetration [50,51]. Adhesins can assemble into hair-like appendages, such as pili and fimbriae, which help parasites bind to receptors on their hosts [52,53]. InterProScan and MEMSAT both predicted an N-terminal signal peptide, and a non-cytoplasmic/extracellular domain for A0A6S8AHQ2. Such features suggest this protein may be extracellular or secreted, raising the intriguing possibility of its involvement in the EN, perhaps performing a role related to EN adhesion or substrate interaction.

The second candidate protein with features suggesting a potential role in the EN was A142921, which Phyre2 suggested bore similarity to type VI secretion system protein (T6SS) (65.2% confidence). T6SS are contractile proteins used by predatory bacteria to inject cytotoxic effector proteins into prey cells [54,55]. MEMSAT predicted that A142921 possesses an N-terminal signal peptide, an extracellular domain, and a transmembrane domain. Thus, A142921 may have a role associated with the EN, perhaps as a secreted or contractile protein.

The third EN candidate protein potentially involved in the EN was A42866. MEMSAT predicted a transmembrane domain and a short extracellular C-terminus. Moreover, GPS-Lipid predicted a palmitoylation site, suggesting potential association between A42866 and a membrane. The highest confidence match suggested by Phyre2 was trafficking protein particle complex subunit 11 (TRAPPC11) (69.1% confidence), a subunit of the TRAPPIII complex involved in vesicular trafficking and autophagy [56–58]. HHPred indicated similarity to BAR protein (64.82% probability), as well as the BAR-domain containing protein endophilin (55.3% probability), which can be involved in modulating/sensing membrane curvature, vesicle membrane fission, and vesicle uncoating [59–62]. These predictions suggest that A42866 may have a similar function related to membrane remodeling or vesicular trafficking.

Two additional candidate proteins (A47522 and A141237) had clearly recognizable structural or functional domains, but ambiguous predicted functions; we wonder whether these 'missing' functions may be bothrosome or EN related. The highest confidence match to A47522 from both Phyre2 and HHPred was "SXP/RAL-2 family protein" (74.9% confidence and 48.81% probability, respectively), which are proteins of unknown function secreted by parasitic nematodes [63]. Supporting the possibility that A47522 might be extracellular, MEMSAT predicted an extracellular N-terminus, followed by a transmembrane domain.

A141237 was associated with the Pfam/InterPro/PANTHER domain "late embryogenesis abundant protein" (LEA, PF02987, IPR004238, PTHR47372:SF49). LEA proteins confer desiccation tolerance in plants and animals [64]. HHPred suggested high confidence similarity to various apolipoproteins, with the highest probability match of apolipoprotein E (98.53% probability). Apolipoprotein E is involved in the binding of cell-surface receptors with lipoproteins [65]. Thus, although an exact function of A141237 will ultimately need to be determined by experiment, these findings hint of a possible extracellular or EN-associated role.

Finally, three candidate proteins (A46070, A442, and A43122) were too divergent for clear bioinformatic inference (other than MEMSAT predicted that all three possess transmembrane and extracellular domains). These enigmatic proteins are particularly interesting candidates for EN-related functions, since the EN itself is unique to labyrinthulomycetes, potentially explaining why deep homology tools were ineffective. Another reason might be related to the fact that Phyre2 and InterProScan indicated they were likely highly disordered proteins. Future experiments are required to resolve their functions.

The candidate proteins identified here provide a starting point for unraveling the molecular basis of EN and bothrosome biology in thraustochytrids. Their predicted features, such as membrane association and extracellular or vesicular trafficking-related roles, align with what we know about the unique structural and functional characteristics of the EN and

bothrosome. While their precise roles remain speculative, these proteins represent prime targets for uncovering evolutionary innovations tied to the emergence of these specialized organelles, and await future experimental validation.

## Conclusions

The transition from swimming zoospores to settled vegetative growth is a critical life-history event in many protists, including thraustochytrids and other heterotrophic stramenopiles, such as oomycetes. In our proteomic analysis of this transition in *A. limacinum*, we demonstrated clear shifts in protein expression: zoospores emphasized motility, signaling, and lipid-based energy mobilization, whereas vegetative cells prioritized nutrient acquisition and anabolic metabolism to support growth and proliferation.

These results align with previous observations, including that *A. limacinum* zoospores can remain motile for days powered by TAG lipid reserves [21]. Although the experiments were done quite differently, our results also align with Dellero et al.'s [66] transcriptional (RNA-seq) study which compared a culture enriched in zoospores with a culture enriched in mono- and multi-nucleated vegetative cells. Among the 312 genes identified as significantly differentially expressed in both studies, there was general agreement that genes, ko- and KOG classes related to motility, signal transduction, peroxisomal beta-oxidation, and lipolysis were strongly upregulated in zoospores, while genes related to DNA replication, transcription, translation, amino acid metabolism, and glycolysis were strongly downregulated (S1G Table). The abundance changes in proteins related to lipid transport and metabolism provide clues to *Aurantiochytium*'s strategy for lipid storage and utilization.

Particularly striking in our results is the indication that ribosomes and several components of protein maturation and turnover machinery may be stockpiled in zoospores. Contrary to the zoospore-enriched genes related to motility, proteasomal, ribosomal, and chaperone protein transcripts were downregulated in the zoospore transcriptome of Dellero et al. [66] (S1G Table), an observation consistent with the stockpiling hypothesis – that is, transcripts of these genes are more abundant in vegetative cells during zoosporogenesis while proteins of these genes are more abundant in mature zoospores. This suggests a preparedness for rapid protein synthesis and turnover, which is crucial for the timely production of enzymes involved in amino acid, coenzyme, and nucleotide production to immediately support growth and development demands post-settlement, and raises the question whether mRNA may also be stockpiled in the zoospore. There were only 26 genes with transcripts more abundant in zoospores but proteins more abundant in vegetative cells (S1G Table), including 6 members of ko03009 (ribosomal biogenesis), suggesting the possibility that some mRNA stockpiling may also occur in zoospores to poise them for the transition to the vegetative cell growth phase.

The identification of proteins with potential roles in the EN or bothrosome opens new avenues for understanding the structural, physiological, and ecological adaptations of labyrinthulomycetes, as well as for studying the evolutionary relationships of the bothrosome and EN to structures in other organisms. Due to the uniqueness of the identified EN/bothrosome candidates, reliable predictions of their precise roles and functions remain elusive. Functional genomic evidence – for example from gene knockout mutants and protein localization studies – is required to confirm if the candidates indeed have EN-related roles, and what they actually do. Such research will illuminate the molecular origins of the EN, a morphological wonder peculiar to the Labyrinthulomycota. Furthermore, it will expand upon our understanding of the ability of eukaryotic membranes and the cytoskeleton to orchestrate complex reticulate structures, a field of study with important implications for human diseases and of potential biotechnological interest.

## Materials and methods

### Zoospore isolation and timing of vegetative cell development

*Aurantiochytrium limacinum* ATCC MYA-1381 [28,67] was grown at 25 °C in a 50 mL liquid culture of GPY media (3% D-glucose, 1.5% peptone, 0.5% yeast extract, 1.8% Instant Ocean sea salt) on an orbital shaker at 100 rpm. After 24 hours, the culture was spread on d-GPY agar plates (2% D-glucose, 1% peptone, 0.5% yeast extract, 1.75% Instant Ocean, 1% agar) and grown at 25 °C for 24 hours. Zoospore release was achieved by flooding the plate with 15 mL of

artificial seawater (ASW: 1.8% Instant Ocean in dH$_2$O, filter sterilized), and waiting 2 hours when zoospore concentration in ASW pipetted off the flooded plate was estimated to be 2.75 million cells/mL using a Neubauer chamber. For microscopy, 500 µL of freshly isolated *Aurantiochytrium limacinum* zoospores were inoculated into 35 mm Nunc™ Glass Bottom Dishes (ThermoFisher Scientific, USA) containing 5 mL sterile A1 media (containing glycerol, maleic acid, and sodium glutamate in artificial seawater; [68]). The petri dish was left undisturbed for 15 minutes to allow the zoospores to settle down and attach to the glass surface. Then the petri dish was placed on the stage of a custom-assembled spinning disk confocal microscope consisting of an automated Zeiss frame, a Yokogawa CSU10 spinning disc, a Ludl stage controlled by a Ludl MAC6000, and an ASI filter turret attached to a Photometrics Prime 95B camera controlled with Metamorph software (version: 7.10.2.240). Settled *A. limacinum* cells were visualized using an oil-immersion Plan Apochromat 63x/1.4 NA DIC objective. Time-lapse imaging, including z-stack, was performed on 5–7 different fields for 12 hours with a 5 minute interval. Zoospore settling and vegetative cell growth were assessed for motility, presence of flagella, cell size, and EN development.

## Proteomics

Zoospores were collected from a 1-day-old petri dish culture on d-GPY agar medium in 15 mL ASW as described above. 1 mL of zoospore solution was inoculated into each of twelve 150 mm diameter Petri dishes containing 35 mL liquid A1 media. Zoospores from the remaining 3 mL were collected by centrifugation (10 minutes, 2,400 x g) for time 0 (T0). After 2, 4, 6, and 8 hours (T2, T4, T6, and T8, respectively), cells were released from three replicate petri dishes using a cell scraper and collected by centrifugation. The biomass of each pellet was measured using a precision balance (Mettler Toledo AL54) and pellets were stored at −80 °C for protein extraction the following day. Each of five time points was represented by 3 biological replicates. Samples at Time 0 (T0) served as a reference point against which changes at subsequent time points (T2, T4, T6, T8) were compared.

The frozen cell pellets were resuspended in 1 mL of lysis buffer (500 µL 1M KCl, 1000 µL 25 mM MgCl$_2$, 500 µL 1M Tris pH 8.3, 0.45% NP40, 0.45% Tween, 100 µL 0.5% SDS, 8210 µL dH$_2$O), and vortexed for 20 minutes at high speed. Then, the mixture was centrifuged at 15,000 xg, 4 °C for 15 minutes. The supernatant was collected, placed on dry ice, and sent for protein analysis by Creative Proteomics (Shirley, NY, USA). Protein concentration was quantified using a bicinchoninic acid (BCA) assay (Thermo Fisher Scientific, USA). 60 µg of protein from each sample were prepared using dissolution buffer, denaturant, and reducing reagent from the BCA kit. Cysteine residues were blocked using cysteine blocking reagent, and the proteins were digested using trypsin solution from bovine pancreas (Promega). Peptides were labeled per replicate using iTRAQ-8plex (Sigma) tags 113, 114, 115, 116, and 117 for time points 0, 2, 4, 6, and 8 respectively. The labeled peptides were combined into one tube per replicate.

Peptide fractionation was performed per replicate with 15 components using HPLC. We used an Ultimate 3000 nano UHPLC system coupled with a Q Exactive mass spectrometer (Thermo Fisher Scientific, USA) with an ESI nanospray source. For iTRAQ-labeled samples, the full scan was performed between 350−1,650 m/z at the resolution 120,000 at 200 Th, and the automatic gain control target for the full scan was set to 3e6. The MS/MS scan was operated in Top 15 mode using the following settings: resolution 30,000 at 200 Th; automatic gain control target 1e5; normalized collision energy at 32%; isolation window of 1.2 Th; charge state exclusion: unassigned, 1, > 6; dynamic exclusion 40 s.

Raw MS files were analyzed separately against the two available predicted proteomes for *Aurantiochytrium limacinum,* from JGI (https://mycocosm.jgi.doe.gov/Aurli1/Aurli1.home.html) [30] and from MMETSP (from the 2020_06 uniprot_trembl release) [69], using Maxquant (1.6.2.14). The parameters were set as follows: protein modifications were carbamidomethylation (C) (fixed), oxidation (M) (variable), itraq-8plex; enzyme specificity was set to trypsin; maximum missed cleavages were set to 2; precursor ion mass tolerance was set to 10 ppm and MS/MS tolerance was 0.6 Da. 3006 of the 14859 (20.2%) JGI-predicted proteins, and 3298 of the 14622 (21.9%) MMETSP-predicted proteins were detected (Maxquant output is available at Dryad DOI https://doi.org/10.5061/dryad.2z34tmpxj). To combine the results into a single

dataset containing all detected and significant proteins, we bidirectionally compared all the predicted MMETSP and JGI proteins using Diamond v2.0.10.148 [70]. 9,673 proteins formed pairs of reciprocal best hits. In addition, 3,356 JGI-predicted proteins with no MMETSP hit,1,886 MMETSP-predicted proteins with no JGI hit, and 1,866 MMETSP plus 3,099 JGI proteins that did not form reciprocal best hit pairs were initially combined (totally 19,880 proteins). After applying additional filtering to remove ambiguous or redundant assignments, the final non-redundant predicted proteome comprised 19,844 proteins. In total, 3,783 unique proteins were detected (S2 Table).

Throughout the text and tables, protein models from the JGI Aurli1 genome annotation are referenced using their original protein ID numbers with a prepended "A" (e.g., A43122). These identifiers can be searched at the JGI Aurli1 gene model portal by removing the "A" prefix and entering the numeric portion (e.g., "43122") in the "Protein ID" field (https://phycocosm.jgi.doe.gov/Aurli1/). In contrast, the MMETSP-derived proteins use UniProt accession numbers (e.g., A0A6S-8DQS1). These accessions follow the UniProt standard format, beginning with "A0A" followed by seven alphanumeric characters. These entries are available from the UniParc archive (https://www.uniprot.org/uniparc).

## Differential abundance analysis

Differential abundance analysis was performed using the Bioconductor package DEP v1.20.0 and following their best practices [71] in parallel for the MMETSP and JGI datasets. For each of the two datasets, we created a summarized experiment object with the time points and replicates using the SummarizedExperiment Bioconductor package. A batch correction was performed using Combat from the package SVA, intensity values were normalized using variance stabilizing transformation (VSN), and limma was used to calculate empirical Bayes moderated t-statistics for time points 2, 4, 6, and 8 with respect to time point 0. The significance thresholds were adjusted to p-value <0.1 and fold-change greater than 1.0001 for upregulation or the inverse $1/1.0001 = 0.9999$ for downregulation. Of the 3006 detected JGI proteins, 394 were significantly differentially expressed for at least one of the later time points when compared to time point 0, and 530 of the 3298 MMETSP proteins were significant. In total, 623 unique proteins were significantly differentially expressed (S2 Table).

To support interpretation, several annotations were added to the dataset. First, depending on the expression level at time point 0, all proteins with above median expression were classified as a set of highly expressed proteins. Second, the datasets were joined to Labyrinthulomycete orthogroup cluster tables collected from JGI (https://phycocosm.jgi.doe.gov/clm/run/Honfer1-comparative-qc.4746;bx5D-n?organismsGroup=labyrinthulomycota) to determine if a protein had homologs in other Labyrinthulomycete species; a protein was considered to be conserved if its orthogroup was present in 3 or more of the 5 genomes. Pfam annotations were added from the JGI proteome (https://phycocosm.jgi.doe.gov/mycocosm/annotations/browser/pfam/summary;I9DIJz?p=Aurli1). Predicted subcellular localizations were obtained for the detected proteins using MultiLoc2 [72]. KEGG assignments were obtained from BlastKOALA [73], GO terms were obtained using OmicsBox/Blast2GO [74], and other domain predictions from Panther [75], ProSite [76], Gene3D [77], and MobiDBLite [78] were acquired using InterProScan v5.55.88 [79].

Potential bothrosome/EN-related candidate proteins were identified as proteins which were upregulated in vegetative cells and did not have close homologs in organisms lacking a bothrosome/EN – that is, outside of Labyrinthulomycota. We used both a BLAST-based method and an orthogroup-based method to identify homologs of detected proteins. The first identified proteins which had BLASTp hits in labyrinthulomycetes, but not other stramenopiles or other organisms. We compared all *A. limacinum* proteins with a stramenopile protein database (all proteins in the Genbank nr database with taxID 33634; downloaded Sept. 27 2022) using BLASTp. We selected the top five hits for each protein based on the lowest e-value and highest percent identity. We grouped the query proteins into four groups: (1) no hits in any stramenopiles (aurli_only), (2) hits only in *Hondaea fermentalgiana* (laby_only), (3) at least 5 better hits in stramenopiles other than *H. fermentalgiana* (stramenopile_not_hondaea), or (4) hits in both *H. fermentalgiana* and other stramenopiles (both). For the last group, we further analyzed the differences between *H. fermentalgiana* and other stramenopiles by subtracting their

respective scores (bitscore, e-value and percent identity). We categorized the bitscore difference distribution between hits to *H. fermentalgiana* vs other stramenopiles into different quantiles (0–85%, 85%−90%, 90%−95%, 95%−97.5% and 97.5%−100%), where proteins with a larger difference were more similar to *H. fermentalgiana* proteins than other stramenopile proteins. The second method identified *A. limacinum* protein homologs by using OrthoFinder v.2.5.5 [80] to cluster sequences from a protist-enriched database [12]. We filtered detected proteins with the following four conditions: (1) the protein was present in at least two Labyrinthulomycota genomes, (2) the protein was not present in more than one non-Labyrinthulomycota genome, and (3) the protein was upregulated in timepoints T4, T6, or T8, and (4) the protein had at least one copy in *Hondaea fermentalgiana*. Using the BLAST or OrthoFinder methods, 18 EN/bothrosome candidate proteins were identified, listed in Table 1. Further bioinformatic analyses were performed in order to hypothesize potential protein functions of the candidates. Two remote protein homology detection tools were used, Phyre2 and HHPred [81,82], using default parameters. Further protein features were predicted using GPS-Lipid, HAMAP, and PsiPred with MEMSAT [83–88], using default parameters, except GPS-Lipid which used the high stringency threshold setting. The output of these remote homology detection searches are summarized in S1E Table.

## Functional enrichment statistics and visualization

Enrichment analyses for KOG classes and ko groups were performed using Fisher's Exact Test per cluster, as well as combining down- and up-regulated clusters. These tests were performed using the hypergeometric function dhyper from the R package stats, and corrected for multiple testing using the Benjamini-Hochberg procedure. Differential enrichment among clusters for gene ontology (GO) terms was conducted with R (v4.4.0) package topGO (v2.56.0) using the weight01 algorithm with Fisher's Exact Test, with a Benjamini–Hochberg correction (5% false discovery rate).

PCA plots, heatmaps, volcano plots, and bar plots for proteins of interest were created with DEP v1.20.0. Venn diagrams were created with ggVennDiagram v1.5.2 [89].

All code is available at https://github.com/phylogenomic/laby_proteomics/tree/main

## Supporting information

**S1 File. Original stills from time-lapse video microscopy used to make Fig 1.** Compressed in zip format.
(ZIP)

**S1 Tables. Supplemental Tables S1A to S1G.** Please see the 'ReadMe' page of the excel file for legends of each table.
(XLSX)

**S2 Table. Proteomic differential expression analysis and annotation spreadsheet.** Please see the 'ReadMe' page of the excel file for explanation of each column in the main data page.
(XLSX)

**S1 Appendix. Supporting figures and legends for all supporting information files.**
(PDF)

## Acknowledgments

{none}

## Author contributions

**Conceptualization:** Anbarasu Karthikaichamy, Joshua S. Rest, Jackie L. Collier.

**Data curation:** Alejandro Gil-Gomez, Ben Leyland, Jackie L. Collier.

**Formal analysis:** Alejandro Gil-Gomez, Ben Leyland, Joshua S. Rest, Jackie L. Collier.

**Funding acquisition:** Anbarasu Karthikaichamy, David Q. Matus, Joshua S. Rest, Jackie L. Collier.

**Investigation:** Anbarasu Karthikaichamy, Rebecca C. Adikes.

**Methodology:** Alejandro Gil-Gomez, Ben Leyland, Anbarasu Karthikaichamy, Rebecca C. Adikes, David Q. Matus, Joshua S. Rest.

**Project administration:** David Q. Matus, Joshua S. Rest, Jackie L. Collier.

**Resources:** David Q. Matus.

**Supervision:** David Q. Matus, Joshua S. Rest, Jackie L. Collier.

**Visualization:** Alejandro Gil-Gomez, Anbarasu Karthikaichamy, Joshua S. Rest.

**Writing – original draft:** Ben Leyland, Joshua S. Rest, Jackie L. Collier.

**Writing – review & editing:** Alejandro Gil-Gomez, Ben Leyland, Anbarasu Karthikaichamy, Rebecca C. Adikes, David Q. Matus, Joshua S. Rest, Jackie L. Collier.

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
