## [Decision Letter · Decision Letter 0]

Dear Dr. Collier,

Thank you for submitting your manuscript to PLOS ONE. After careful consideration, we feel that it has merit but does not fully meet PLOS ONE’s publication criteria as it currently stands. Therefore, we invite you to submit a revised version of the manuscript that addresses the points raised during the review process.

Your study presents an intriguing proteomic analysis of the transition from zoospores to vegetative cells in Aurantiochytrium limacinum. It provides valuable insights into the molecular processes underpinning the formation of ectoplasmic networks and bothrosomes, as well as broad cellular changes during this transition. While the study is the first to focus on these unique structures in thraustochytrids, some aspects remain preliminary. The reviewers appreciated the significance of your findings but noted several issues requiring clarification to enhance the manuscript's impact.

We look forward to receiving your revised manuscript.

Kind regards,

Alberto Amato

Academic Editor

PLOS ONE

Journal Requirements:

Reviewers' comments:

Reviewer's Responses to Questions

**Comments to the Author**

1. Is the manuscript technically sound, and do the data support the conclusions?

Reviewer #1: Yes

Reviewer #2: Partly

Reviewer #3: Yes

2. Has the statistical analysis been performed appropriately and rigorously?

Reviewer #1: Yes

Reviewer #2: I Don't Know

Reviewer #3: Yes

3. Have the authors made all data underlying the findings in their manuscript fully available?

Reviewer #1: Yes

Reviewer #2: Yes

Reviewer #3: Yes

4. Is the manuscript presented in an intelligible fashion and written in standard English?

Reviewer #1: Yes

Reviewer #2: Yes

Reviewer #3: Yes

Reviewer #1: The manuscript by Alejandro Gil-Gomez et al describes proteomic analysis of the zoospore-to-vegetative cell transition of the stramenopile Aurantiochytrium limacinum. The authors first performed time-course proteomics analysis of the transition from zoospores to vegetative cells and identified the differential expression of proteins in broad cellular changes, such as shifts in motility, signaling, and metabolism. They also focused on the formation of the ectoplasmic network and bothrosome and identified candidate proteins involved in their formation. Even though this study did not contain experimental data for the function of the identified proteins, they discussed them well, and their interpretations were sound. Furthermore, this is the first proteomic study to focus on the transition from zoospores to vegetative cells in this unique microorganism, and this study contributes to our understanding of thraustochytrids physiology. Although this manuscript contains some important observations, it has several issues that need to be addressed before acceptance of PLOS One.

Comments:

Figure 5 shows that almost all proteins associated with proteasomes were downregulated during the transition from zoospores to vegetative cells. This point needs to be clearly explained.

The reviewer could not access the protein information with ID numbers, including letters such as A0A6S8DQS1.

Table 1, definition of ID should be included. Is that a JGI database number? Definition of ? for A43122, A442, and A46070 should be included.

S8C, S8G, S8H Figure, the figure legend does not match the figure.

Line 669, “Mar Rundsch” might be “Mar Biotechnol”.

Reviewer #2: This manuscript investigates the molecular components of the ectoplasmic network (EN) and the bothrosome in thraustochytrids, using a time-course proteomic analysis to compare motile zoospores and substrate-attached vegetative cells. The EN, a unique structure formed upon attachment, is associated with the cytoplasmic bothrosome; however, the molecular details of these structures remain largely unknown.

The authors first identify broad cellular changes between zoospores and vegetative cells, such as increased expression of proteins involved in β-oxidation and anabolic metabolism, respectively. While these findings are consistent with previous studies, they do not provide novel insights.

The main contribution of the study lies in the identification of candidate proteins that are specifically expressed in vegetative cells and appear to be unique to thraustochytrids. These proteins are proposed as potential components of the EN or bothrosome. Although this list may serve as a valuable resource for future research, the study remains preliminary in its current form.

To strengthen the manuscript, I recommend the authors include experimental validation of the localization of the candidate proteins within the EN or bothrosome structures. Techniques such as in situ hybridization or immunostaining using antibodies against selected target proteins would provide more direct evidence and significantly enhance the impact and credibility of the study.

Reviewer #3: In this work, the authors present a very interesting proteomic analysis of the zoospore-to-vegetative cells transition in the model thraustochytrid Aurantiochytrium limacinum. The manuscript is well written and shed some light on the elusive formation of ectoplasmic nets and bothrosome in these unique organisms. The principal limitation of this work is that it might not rely enough on other studies to comment the findings. For instance, adequation with RNAseq results obtained in ref. 66 is indicated, but would deserve much broader discussion, e.g. by comparing RNA and Protein data during time course. Also, it would be interesting to know is there is any data about the potential EN candidates (or related) in this referenced study or any other omics work and, if not, discuss the reasons why, e.g. in relation to experimental conditions, etc.

A few minor points:

- Although the manuscript generally follows the unit rules of the v9 (2019) BIPM's brochure, there are some discrepancies. The brochure says "The internationally recognized symbol % (percent) may be used with the SI. When it is

used, a space separates the number and the symbol %." Please correct this.

- In the protein extraction process, a centrifugation step is only provided with an angular speed (rpm) which is not a suitable unit. If possible, please provide an acceleration (xg) as correctly done earlier in the manuscript.

**Do you want your identity to be public for this peer review?** For information about this choice, including consent withdrawal, please see our Privacy Policy

Reviewer #1: No

Reviewer #2: No

Reviewer #3: No

---

## [Author Response · Author response to Decision Letter 1]

29 May 2025

This response also uploaded with the revised manuscript.

Below, we respond point-by-point to the reviewers’ comments. We thank the reviewers for their efforts, which helped us improve this manuscript, and hope they will find our consideration of their comments satisfactory.

Reviewer #1: The manuscript by Alejandro Gil-Gomez et al describes proteomic analysis of the zoospore-to-vegetative cell transition of the stramenopile Aurantiochytrium limacinum. The authors first performed time-course proteomics analysis of the transition from zoospores to vegetative cells and identified the differential expression of proteins in broad cellular changes, such as shifts in motility, signaling, and metabolism. They also focused on the formation of the ectoplasmic network and bothrosome and identified candidate proteins involved in their formation. Even though this study did not contain experimental data for the function of the identified proteins, they discussed them well, and their interpretations were sound. Furthermore, this is the first proteomic study to focus on the transition from zoospores to vegetative cells in this unique microorganism, and this study contributes to our understanding of thraustochytrids physiology. Although this manuscript contains some important observations, it has several issues that need to be addressed before acceptance of PLOS One.

Comments:

Figure 5 shows that almost all proteins associated with proteasomes were downregulated during the transition from zoospores to vegetative cells. This point needs to be clearly explained.

>Indeed, as shown in greater detail in Figure S8B, half of the detected proteasome core proteins (7 of 14) and about a third of the detected proteasome regulatory proteins (7 of 20) were relatively more abundant in zoospores. In the original manuscript, this is described on lines 259, 265, 270, and we agree was not adequately discussed/interpreted. We have expanded on that in the Conclusion, where we discuss the interpretation that protein biosynthetic and turnover machinery is stockpiled in zoospores, and now include our observations that most of the proteasomal proteins show the opposite behavior in Dellero et al.’s transcriptomic analysis: that is, they are enriched in zoospores at the protein level but depleted at the transcript level. See response to Review 3 for more detail on the analysis comparing our proteomic results to Dellero's RNA-seq experiment.

The reviewer could not access the protein information with ID numbers, including letters such as A0A6S8DQS1.

>We have added a paragraph to the end of the ‘Proteomics’ section of Materials and Methods detailing how each predicted protein can be accessed in the appropriate database. We hope this addresses the reviewer’s difficulty.

Table 1, definition of ID should be included. Is that a JGI database number? Definition of ? for A43122, A442, and A46070 should be included.

>This issue is also clarified by the above described additional text.

S8C, S8G, S8H Figure, the figure legend does not match the figure.

>We have carefully reviewed Figures S8C, S8G, and S8H, their corresponding legends, and all references to them in the main text. At this time, we are unable to identify a discrepancy between the figure legends and the figures themselves. We respectfully ask the reviewer, if possible, to clarify the specific mismatch they observed so that we can address it appropriately. We hope that upon a second review, the figures and legends will be found to be in alignment.

Line 669, “Mar Rundsch” might be “Mar Biotechnol”.

>Thank you for pointing this out. The citation has now been corrected to Mar Biotechnol in the revised manuscript.

Reviewer #2: This manuscript investigates the molecular components of the ectoplasmic network (EN) and the bothrosome in thraustochytrids, using a time-course proteomic analysis to compare motile zoospores and substrate-attached vegetative cells. The EN, a unique structure formed upon attachment, is associated with the cytoplasmic bothrosome; however, the molecular details of these structures remain largely unknown.

The authors first identify broad cellular changes between zoospores and vegetative cells, such as increased expression of proteins involved in β-oxidation and anabolic metabolism, respectively. While these findings are consistent with previous studies, they do not provide novel insights.

The main contribution of the study lies in the identification of candidate proteins that are specifically expressed in vegetative cells and appear to be unique to thraustochytrids. These proteins are proposed as potential components of the EN or bothrosome. Although this list may serve as a valuable resource for future research, the study remains preliminary in its current form.

To strengthen the manuscript, I recommend the authors include experimental validation of the localization of the candidate proteins within the EN or bothrosome structures. Techniques such as in situ hybridization or immunostaining using antibodies against selected target proteins would provide more direct evidence and significantly enhance the impact and credibility of the study.

>We appreciate the reviewer’s suggestion to include direct localization evidence for candidate EN or bothrosome proteins. We fully agree that such experiments (e.g., in situ hybridization or immunostaining) would provide a powerful complement to our proteomic results. However, we respectfully note that the generation of the required tools (custom antibodies, probe sets, or transformation constructs, etc.) is a significant and ongoing undertaking. While work towards localization studies is indeed underway in our lab, this is a substantial long-term project that is beyond reasonable inclusion in the present manuscript.

We believe that the current study already makes a substantial contribution by providing the first global proteomic map of the zoospore-to-vegetative transition and a prioritized set of candidate proteins associated with EN/bothrosome formation. In addition, in response to reviewer 3, we now incorporate a more systematic analysis of Dellero et al. (2020). Some key proteins enriched in zoospores at the proteomic level are found to be downregulated at the transcript level in Dellero’s zoospore-rich cultures. This pattern is consistent with stockpiling of proteins in zoospores (see response to reviewer 3; supporting proteins stored in the zoospore for rapid deployment upon cell attachment and EN formation). We also now note in S1E Table how expression of proteins from our candidate list differed between zoospore-rich and zoospore-poor cultures in the Dellero et al. RNA-seq study.

To acknowledge the importance of future validation, we have added mention to the Conclusions of the sorts of follow-up functional genomic work that is needed. We hope the reviewer agrees that the present study already lays a valuable foundation for these next steps.

Reviewer #3: In this work, the authors present a very interesting proteomic analysis of the zoospore-to-vegetative cells transition in the model thraustochytrid Aurantiochytrium limacinum. The manuscript is well written and shed some light on the elusive formation of ectoplasmic nets and bothrosome in these unique organisms. The principal limitation of this work is that it might not rely enough on other studies to comment the findings. For instance, adequation with RNAseq results obtained in ref. 66 is indicated, but would deserve much broader discussion, e.g. by comparing RNA and Protein data during time course. Also, it would be interesting to know is there is any data about the potential EN candidates (or related) in this referenced study or any other omics work and, if not, discuss the reasons why, e.g. in relation to experimental conditions, etc.

>We have limited our comparison to the Dellero et al. 2020 (ref 66) RNAseq study because the experimental conditions used there were substantially different from ours. Specifically, Dellero et al. compared 1-day-old cultures grown in R (relatively rich) vs P (relatively poor) liquid media on an orbital shaker at 100 rpm, which contained very few (<3%) vs mostly (85%) zoospores, respectively (Dellero et al. 2020, Table 1). We do not know which, if either, of these cultures were producing EN. Possibly the R cultures were, but they were started not from zoospores but from a stationary phase culture in rich medium which did not contain zoospores, and the single 1-day timepoint does not overlap our (shorter) timecourse. Additionally, the growth media were quite different. Our vegetative cells had glycerol and glutamate as organic resources, while Dellero et al. used a more typical growth medium with glucose and yeast extract. Thus, while we are confident that general features of zoospore gene expression can be compared between their study and ours, we don’t expect other features of our study - particularly the time-course of vegetative cell development, including EN production - to be evident in the Dellero et al. 2020 work. We will also note that there is not generally a very tight correlation between RNAseq and proteomics studies (i.e. correlation coefficients typically low: ~0.16 across strains, ~0.4 under nitrogen limitation; overlap between DEGs and DEPs can vary widely, e.g. 3-30%, from Marcel et al. 2024 PNAS, Yu et al. 2020 Nat Commun, Li et al. 2019 Biotechnol Biofuels). Despite these cautions, we performed a detailed comparison and have added to the manuscript a supplemental table (S1G), a column to S1E, and new text to the Conclusions to report what we found.

>We have not been able to find any information on potential annotations or functions of the candidate genes beyond what we provide in the manuscript. We found that 5 of them were ‘downregulated’ in Dellero et al. 2020 (ref 66), meaning that their transcripts were relatively less abundant in zoospore-rich cultures (3 could not be compared because the proteins were absent in the JGI proteome, and 10 were either not detected or not significant in Dellero et al.). However, it is not clear what results would bolster or weaken the potential for these as being involved in the EN/bothrosome, because we do not know the status of the Dellero experiments regarding production of EN/bothrosome. While the zoospore-enriched culture may be comparable to our zoospore samples, it is often said that cultures grown shaking in rich medium do not produce EN - it seems possible that what this means is that it is not easy to identify the EN that is produced in such cultures. We have added a column containing this information to Supplemental Table S1E.

A few minor points:

- Although the manuscript generally follows the unit rules of the v9 (2019) BIPM's brochure, there are some discrepancies. The brochure says "The internationally recognized symbol % (percent) may be used with the SI. When it is

used, a space separates the number and the symbol %." Please correct this.

>Thank you for pointing this out; we have fixed this.

- In the protein extraction process, a centrifugation step is only provided with an angular speed (rpm) which is not a suitable unit. If possible, please provide an acceleration (xg) as correctly done earlier in the manuscript.

>Thank you for pointing this out; we now provide the acceleration.

---

## [Editor Report · Decision Letter 1]

Proteome remodeling in the zoospore-to-vegetative cell transition of the stramenopile Aurantiochytrium limacinum reveals candidate ectoplasmic network proteins

PONE-D-25-16410R1

Dear Dr. Collier,

We’re pleased to inform you that your manuscript has been judged scientifically suitable for publication and will be formally accepted for publication once it meets all outstanding technical requirements.

Kind regards,

Alberto Amato

Academic Editor

PLOS ONE
---

## [Editor Report · Acceptance letter]

PONE-D-25-16410R1

PLOS ONE

Dear Dr. Collier,

I'm pleased to inform you that your manuscript has been deemed suitable for publication in PLOS ONE. Congratulations! Your manuscript is now being handed over to our production team.

Kind regards,

on behalf of

Dr. Alberto Amato

Academic Editor

PLOS ONE